# Waste from Artichoke Processing Industry: Reuse in Bread-Making and Evaluation of the Physico-Chemical Characteristics of the Final Product

**DOI:** 10.3390/plants11243409

**Published:** 2022-12-07

**Authors:** Michele Canale, Alfio Spina, Carmine Summo, Maria Concetta Strano, Michele Bizzini, Maria Allegra, Rosalia Sanfilippo, Margherita Amenta, Antonella Pasqualone

**Affiliations:** 1Research Centre for Cereal and Industrial Crops, Council for Agricultural Research and Economics (CREA), Corso Savoia, 190, 95024 Acireale, Italy; 2Department of Soil, Plant and Food Sciences, University of Bari Aldo Moro, Via Amendola, 165/A, 70126 Bari, Italy; 3Research Centre for Olive, Fruit and Citrus Crops, Council for Agricultural Research and Economics (CREA), Corso Savoia, 190, 95024 Acireale, Italy; 4Stazione Consorziale Sperimentale di Granicoltura per la Sicilia, Santo Pietro, 95041 Caltagirone, Italy

**Keywords:** durum wheat, artichoke, quality characteristics, leavening test, functional bread, metabolic diseases, sustainability, upcycling, food waste, absorption capacity

## Abstract

A relevant amount of waste is produced in the canning industry of globe artichoke. This study proposes to use flours of artichoke waste (stems and bracts) in durum wheat bread-making, replacing the re-milled durum wheat semolina at increasing levels (5, 7.5 and 10 g/100 g). No study had evaluated this type of enrichment in durum wheat bread, widespread in the same area where artichoke waste is mostly produced. The replacement had a visible effect on the flour color, increasing *a** and reducing *b** and *L**, and this was reflected in the color of bread crumb. The water absorption determined by farinography, dough development time and dough stability increased as the level of replacement increased (up to 71.2 g/100 g, 7.3 min and 18.4 min, respectively). The mixograph peak height and mixing time increased compared to control. The alveograph W decreased, while the P/L ratio increased. The artichoke waste-enriched breads had a lower volume (as low as 1.37 cm^3^/g) and were harder than control, but they did not show relevant moisture losses during five days of storage. The obtained data show therefore an interesting potential of artichoke waste flours in bread-making, but further investigations are needed for achieving improved quality features.

## 1. Introduction

Globe artichoke (*Cynara cardunculus* subsp. *Scolymus* (L.) Hayek) is the basic ingredient of several domestic and industrial food preparations typical of the Mediterranean foodscape. In addition to the gastronomic value, artichoke has recognized pharmaceutical properties due to the presence of bioactive compounds and antioxidants [1], not to mention its content of minerals and fibers, insoluble and soluble. In detail, the medicinal properties of artichoke are specifically linked to high levels of polyphenols, such as cynarine, together with its biosynthetic precursor chlorogenic acid, and to the presence of inulin, a hypoglycemic oligosaccharide with a positive effect on the intestinal microflora [2]. Artichoke leaf extracts also have a known hepatoprotective effect and are able to decrease cholesterol blood levels [3,4,5].

When the artichokes are harvested, the roots, the leaves and most of the stems are cut, generating numerous scraps that are normally left in the field. The amount of this waste fluctuates between 58.5 and 69% [6]. The artichoke processing industries (mostly canning) generate additional waste and by-products, which at most are used as animal feed. First, the blackened, wrinkled parts of the plant, or the cold-damaged ones, are eliminated. In addition to an initial sorting, a scrap follows due to the calibration and turning of the flower heads (to retain the “heart”), where all the external bracts, the upper part of the internal bracts and the final part of the stem are eliminated. The edible parts of the artichoke, indeed, are only the heart, the internal bracts and the initial part of the stem.

In recent years, there has been a change in the cultural attitude of consumers, with increased attention to issues regarding sustainability and a greater propensity to change lifestyle to go in this direction. This newfound ecological consciousness is reflected in business decisions to shape more sustainable food production systems. The circular economy has become a hot topic and the reuse and upcycling of food industry waste has spurred many research studies [7,8,9].

The use of artichoke waste has been proposed to prepare cakes [10] and pasta [11,12] with potentially functional properties mainly due to the presence of inulin and bioactive compounds. The same waste has been proposed for the production of cheese [13,14,15], in this case exploiting the coagulating properties of artichoke due to the presence of proteases such as cardosin/cyprosin or cynarase [16,17]. Some studies have also been conducted on the use of artichoke fiber in bread-making [18,19], generally considering the production of soft wheat bread.

Among the numerous types of existing breads, in the Mediterranean area—which is the same area where most of the artichoke waste is produced—durum wheat bread is traditionally consumed, prepared with re-milled semolina [20,21,22]. The latter is characterized by smaller particle size and higher hydration rate than semolina for pasta-making and is obtained with an extra milling step compared to semolina [23]. In this perspective, the purpose of the study was to prepare flour from artichoke processing waste (stems and bracts) and evaluate the effect of its addition at increasing levels (5, 7.5 and 10 g/100 g) on the quality of durum wheat bread, with the final aim of increasing the health value of the end-product.

## 2. Results and Discussion

### Characteristics of Flours

The fresh stems of artichoke had a moisture content of 67 g/100 g, whereas the bracts were less moist (57 g/100 g) (Table 1). These values were in agreement with Borsini et al. [24,25]. As for the flours, prepared by oven-drying the stems and bracts at 40 °C for 48 h and subsequently milling them, the moisture content of stem flour (FAS) accounted for 6 g/100 g while the flour of bracts (FAB) had a lower moisture content (4 g/100 g). The 1:1 mix of FAB and FAS (FAM) had an intermediate moisture content. All flours had a moisture content far below 14.5 g/100 g, which is considered a safe level imposed by current rules for other flours, such as wheat flour [26].

Flours from artichoke waste are shown in Figure 1. Those from bracts had a greenish tone, while the flours prepared from stems had a red-brown color, and an intermediate color characterized the mixes of bracts:stems 1:1. Therefore, the colorimetric analysis showed the lowest value of *a** (related to redness) in bract flour (FAB-100%), and the highest in stem flour (FAS-100%), which also showed the highest brown index (100 − *L**) (Table 2). The value of *b** (yellowness) was significantly lower in FAB. The content of reducing sugars has been reported to be higher in the artichoke stem (20.7 g/100 of dry material) than in the bracts (10.4 g/100 of dry material) [27]. Therefore, during oven-drying FAS was probably subjected to a more intense Maillard reaction, becoming browner than FAB.

The flours prepared from artichoke stems and bracts were used to replace re-milled durum wheat semolina at increasing levels (5, 7.5 and 10 g/100 g), for preparing artichoke waste-enriched durum wheat bread. This replacement had a visible effect on the flour appearance, with the same trend observed in 100% artichoke flours. The addition of artichoke flours, especially FAS, increased the *a** and brown index, while having a lowering effect on *b** (Table 2). As for the color parameters of re-milled semolina, they agreed with the usual values for this kind of flour, with the exception of *b**, which was slightly lower [23,28].

Compared to pure re-milled semolina, the water binding capacity (WBC) significantly increased when FAS and FAB were added at 10% replacement level (Figure 2), with the highest value ascertained in FAS, accounting for 1.98 g water/g flour. Artichoke bracts and stems are known to contain relevant amounts of dietary fiber, whose hygroscopic properties explain the increase in WBC. Among the soluble fibers, inulin is in the range 5–8 g/110 g [15,29], with a slightly higher content in stems (7.7 g/100 g) than bracts (6.5 g/100 g) [30]. The ability of inulin to increase water absorption has been reported in several food systems such as pasta dough and ice cream [30,31]. The oil binding capacity (OBC) did not show statistically significant variations after the addition of artichoke waste flours, and was higher than WBC, in agreement with Boubaker et al. [18]. Information about the capacity to absorb water and oil is useful, in addition to defining the correct dosage of water and oil during the preparation of food products, to estimate the ability of retaining water during storage of bakery products, or the capacity to reduce fat losses during processing [32].

As already shown by WBC data, the water absorption determined by farinography (Table 3) also significantly increased as the level of replacement of re-milled semolina increased, due to fibers provided by the artichoke flours. This increase was particularly relevant for FAM and FAS. When these flours were added at the dosage of 10%, the farinograph water absorption reached amounts of 71.2 and 70.2 g/100 g, for FAM and FAS, respectively. A lower increase was observed by adding FAB, without significant variations as the level of replacement increased. The increase in water absorption, due to the ability of fibers to establish hydrogen bonds with water, is of practical interest because it leads to increased dough yield [33,34,35].

The dough development time generally increased with the addition of artichoke flours, again due to the increase in fiber content which slowed down the formation of the dough. Re-milled semolina had a dough development time of 1.8 min, while for FAS-10% 7.3 min were needed. Bran fibers disrupt the continuous protein–starch matrix, negatively affecting dough development and leading to poor visco-elastic properties [36]. The stability of the dough increased after the addition of artichoke flours due to the stiffening effect of fibers, but the softening degree at overmixing also tended to increase. The inulin component of fiber has been reported to increase dough development time and dough stability [37,38,39]. Other authors, who incorporated fiber concentrate from artichoke stem into wheat dough, reported an increase in farinograph stability [18]. The farinograph data of re-milled semolina agreed with results reported in previous studies and were in the normal range for this category of flour [28,40].

The mixograph analysis measures and records the resistance of a dough to mixing. This analysis helps in studying the effect of ingredients on dough mixing properties. The peak height indicates the maximum dough consistency and is an indicator of gluten strength, related to the quality of bread [41]. This peak depends on the absorption of water and protein content, and in wheat refined flour is related to dough strength [42]. A general increase in peak height was observed after the addition of artichoke flours, except for FAM at the higher percentages (Table 4). Presumably, peak height increased due to the greater water absorption induced by fibers, although a positive effect of inulin, capable of partly counteracting the negative effect of other fibers on gluten strength, cannot be excluded. Indeed, it has been reported that inulin can enhance the mechanical properties of the gluten network and improve the dough resistance to mixing [37] thanks to its hydrocolloid properties [43]. Mixing time (to peak) is the time needed to reach peak height, also named dough development time. The addition of artichoke flour caused a significant increase in the mixing time with respect to the control, except for FAM-5%. Overall, the observed values were similar to the mixing times usually ascertained in wheat flours [41].

To achieve an optimal bread development, strong gluten and balanced visco-elastic properties—which can be analyzed by an alveograph—are fundamental. The latter are expressed as tenacity to extensibility ratio (P/L) which, while in soft wheat flours should range from 0.4 to 0.8, in durum wheat re-milled semolina can be as high as 2.5, as ascertained in previous studies [23,28]. Re-milled semolina, indeed, typically has a tenacious gluten. Both the P/L and the dough strength (W) of re-milled semolina agreed with the usual values for this kind of flour [23,28] (Table 4).

The dough strength was negatively affected by the decrease in gluten content after the addition of artichoke flours, which do not contribute gluten. Therefore, the replacement of re-milled semolina with artichoke flours caused a significant decrease in W, irrespective of the type of flour, directly related to the percentage of replacement. On the contrary, P/L ratio increased. The dietary fibers compete with proteins in absorbing water and therefore, since the alveograph protocol involves a constant water amount, the dough was markedly stiffened. In addition, fibers are known to interfere with the optimal formation of the dough, affecting the visco-elastic properties of gluten, as found by Liu et al. [44]. These results suggest that loaves with more compact crumbs and smaller volumes than the control would be obtained when artichoke four is added. P/L, in fact, is known to be negatively correlated with bread specific volume [23].

The leavening test shows that the addition of artichoke waste flours enhanced the fermentation compared to control (Figure 3). The pure re-milled semolina fermented very gradually, while the doughs enriched with artichoke waste flours had a higher leavening rate, reaching high volumes in very short time, likely due to the presence of fermentable dietary fiber, including inulin. On the other hand, the enriched doughs could not withstand prolonged leavening times.

The colorimetric parameters of bread crumb and crust are reported in Table 5. By adding artichoke waste flours, the color of the crumb became brown, similar to that of whole wheat bread (Figure 4), with a brown index reaching two times higher values than the control. The *a** value significantly increased, irrespective of the type of artichoke waste added—bracts, stems or their mix—and of the replacement level, while *b** markedly decreased, with a greater decrease at 10% substitution level. The tendency of the crumb to become significantly darker with the increase in the percentage of artichoke waste flour added was in agreement with the findings of Frutos et al. [19], who enriched soft wheat bread with artichoke fiber. The crust, whose color is essentially due to the Maillard reaction [45], showed a slightly grayish tone when the flours of artichoke wastes were added, but still remained very similar to the control. The value of *a** decreased, but with significant differences only in the case of FAB. There was no significant difference in the brown index and in the *b** value of crust, compared to control.

The physical characteristics of breads are reported in Table 6. Compared to control, the volume and the specific volume of the bread significantly decreased as the replacement percentage of re-milled semolina increased, in agreement with the alveograph data. Among the enriched breads, the highest volume was recorded in the breads prepared with FAS-5% and FAS-7.5%. The progressive reduction of the specific volume is typical of breads in which increasing quantities of fiber are added [46,47,48]. Dietary fibers, indeed, are known to have a detrimental effect on bread volume because they disrupt the gluten network [49] by diluting the gluten-forming proteins [50] and reducing the amount of water available for gluten development [51]. The result is a weakened gluten, which has a reduced ability to retain fermentation gas and ensure a voluminous loaf, as was shown by the results of the leavening test (Figure 3). Probably, a shorter leavening time would allow improvement in the volume of artichoke-enriched breads. Additionally, an increase in the hydration rate, reported to allow the formation of larger pores, could help in obtaining a more open crumb structure [52]. The internal appearance of the loaves was regular in all the examined samples, although the addition of waste artichoke flours caused a slight decrease in the crumb pore size as the amount of added flour increased, due to the mentioned reduced ability to hold carbon dioxide produced during leavening. This decrease was statistically significant only for FAB added at the highest dosage (10%). Bread enriched with other types of fibrous waste, such as brewer’s spent grain or pumpkin pulp (sometimes treated as waste material after taking the seeds), was reported to have smaller and more compact pores (higher porosity score), and smaller bread volume [53,54]. In addition, other authors reported that bread with added inulin of up to 4% also showed smaller crumb porosity [55]. The presence of fibers, highly hygroscopic, also caused a weight increase compared to the control, as found by Fendri et al. [56] using fibrous matrices other than those of artichoke. Artichoke flours therefore can increase the bread yield, thanks to their high WBC.

The moisture content and hardness of the examined breads were studied over 5 days, recording their variations (Table 7). The decrease in moisture was significantly greater and more rapid in bread prepared with pure re-milled semolina than in breads with added FAB, FAS and FAM. The fiber of artichoke flours, indeed, counteracted water loss, in fact the moisture content of the control bread fell below 30% (specifically, it was equal to 27.3 g/100 g) 2 days after baking, while that of the breads enriched with FAB, FAS and FAM, on average, had a moisture loss of 2% only, while maintaining a moisture content higher than 30%. As for the hardness, among the enriched breads FAB-10% was the hardest, while FAS-10% was the softest, the latter without significant differences to control bread. In the case of the control bread, a significant hardening was observed in the first two days after baking, due to starch retrogradation and decrease in moisture [57], however, no significant variation was observed in the successive two days. So, control bread remained the softest sample at the end of the storage period. On the contrary, with the exception of FAB breads, which did not vary significantly in the first two days after baking, all the other enriched breads significantly hardened two days after baking, and further hardened at the end of storage, reaching very high values. Similar findings have been found by other authors [58] when adding fibrous materials. Probably, the hard texture of the artichoke fiber itself and the gluten-diluting effect exhibited by the artichoke flour, the latter allowing the starch to gelatinize and retrograde more freely, caused the more rapid hardening observed in the artichoke-enriched breads.

## 3. Materials and Methods

### 3.1. Preparation of Flour of Artichoke Stems and Bracts

The artichokes cv. Violetto ramacchese were harvested in the last ten days of March 2022 and acquired on March 31, 2022. Harvesting time was chosen to ensure full ripening, which for this cultivar means purple bracts with grayish-green streaks and a 25 to 50 cm long stem having green color without woody formations. The artichokes (at least thirty) were harvested randomly in the fields of the “Violetto Ramacchese” agricultural cooperative, based in Contrada Pietrosa, Ramacca, Italy (GPS coordinates: Latitude N 37°38.5265, Longitude E 14°69.3430). The thirty artichokes collected were then subjected in bulk to the same procedure as in the canning industries. The stems and bracts of artichokes were separated from the hearts to simulate the waste produced by the artichoke canning. Only the external bracts were taken, about 20–25 for each artichoke. The stems were cut into 1.5 cm long pieces. Bracts and stems, independently processed, were placed in aluminum trays and dried at 40 ± 5 °C for about 48 h by using a forced convection drying chamber (Memmert, Milano, Italy), until moisture content was in the range 4–6 g/100 g [59]. After drying, the stems and bracts were coarsely chopped by using a cutter (Imetec, Bergamo, Italy), then milled with a Cyclotec type 120 mill (Perten, Huddinge, Sweden) equipped with a 500 μm sieve. Flour mixes were then prepared by manually mixing stem flour and bract flour in a 1:1 ratio.

### 3.2. Preparation of Breads

The experimental breads were prepared according to the indications given in the AACC method 10.10.02 [60], modified. The formulation of each type of bread is reported in Table 8. The enrichment with artichoke flour did not exceed 10% because after preliminary tests carried out at 50% and 20% it was impossible to perform farinograph analysis due to insufficient gluten network formation. Moreover, other works recommend levels not exceeding 10% for similar vegetable materials [52], and even lower levels for very fibrous materials [61].

Re-milled semolina (Cooperativa Agricola “Valle del Dittaino”, Enna, Italy), pure or partly replaced by flour of stems and bracts, yeast (Lievital, Lesaffre Italia, Parma, Italy), shortening (Gioia, Unigrà srl, Ravenna, Italy), sodium chloride (Italkali, Palermo, Italy), sugar (Decò, Ravenna, Italy), ascorbic acid (Farmitalia, Catania, Italy) and tap water were mixed in an experimental kneader (National Manufacturing Co., Lincoln, NE, USA) at 25 °C for 4 min. The dough was leavened in a thermostatic chamber (Giorik, Sedico, Italy), equipped with a steam humidifier (SD/SD series, Carel, Brugine, Italy), at 30–32 °C, 80–82% RH for 2.35 h, then was poured into metal pans (7 cm width, 12.5 cm length) and leavened again for 50 min at 30–32° C. This was followed by baking in an electric oven (Giorik, Sedico, Italy) for 18 min at 218 ± 5 °C. Two bread-making trials were carried out.

### 3.3. Determination of Moisture Content

Moisture content of fresh stems and bracts, flours and breads was determined by oven-drying (Memmert, Milano, Italy) at 103 °C until constant weight, according to the AOAC method 935.25 [62]. The analyses were carried out in triplicate.

### 3.4. Water Binding Capacity and Oil Binding Capacity

The water binding capacity and oil binding capacity were determined as in Kahraman et al. [63] and Du et al. [64], respectively. An amount of 2 g of flour was added to 24 mL of distilled water or sunflower oil and kept under shaking at 20 °C for 60 min. Following a centrifugation at 4200 rpm × 30 min (Heraeus Multifuge X3 FR, Thermo scientific, Waltham, Massachusetts), the solid residue was weighed. The analyses were carried out in triplicate.

### 3.5. Color Determination

The CR 200 colorimeter (Minolta, Osaka, Japan) was used for color evaluation. The CIELab colorimetric model was adopted by expressing the results according to the coordinates *L**, *a** (indicating the change from green to red) and *b** (indicating the change from blue to yellow). The brown index (100 − *L**) was then calculated, which indicates the tendency to darken ranging from 0 to 100. The analyses were carried out in triplicate.

### 3.6. Farinograph, Mixograph and Alveograph Analyses

A farinograph (Brabender, Duisburg, Germany) was used to determine the water absorption capacity, dough development time, dough stability and degree of softening according to the AACC method 54–21 [60]. An alveograph (Tripette et Renaud, Chopin Technologies, Villeneuve-la-Garenne, France) equipped with the Alveolink software (Tripette et Renaud, Chopin Technologies, Villeneuve-la-Garenne, France) was used to determine the dough strength (W), and tenacity/extensibility ratio (P/L) according to the UNI 10453 method [65]. The mixograph analysis was carried out according to the AACC-approved method 54–40.02 [60]. A mixograph (National Manufacturing Co., Lincoln, NE, USA) was used, equipped with a 10 g bowl, at the mixing speed of 88 rpm and test duration of 10 min. Water amounts to be added to each sample were those determined by the farinograph. The analyses were carried out in triplicate.

### 3.7. Leavening Test

For the preparation of the dough, according to Miś et al. [66], re-milled semolina, pure or partially replaced with increasing percentages (5.0, 7.5, 10%) of FAB, FAS or FAM, and 3 g/100 g dehydrated yeast (Pizza bella alta Paneangeli, Cameo S.p.a., Desenzano del Garda, Italy), preliminarily dissolved in an amount of distilled water at 35 °C corresponding to the farinograph absorption of flour at 500 B.U., were kneaded for 5 min. Subsequently, 25 g of dough was placed in 250 mL graduated cylinders previously oiled to prevent sticking, then the dough was pressed and the starting volume was recorded [67]. The cylinders were kept at a temperature of 30 ± 2 °C in an incubator (Memmert, Milano, Italy) and the percentage of volume increase was determined every 10 min, until two successive equal measurements were recorded [68]. The analyses were carried out in triplicate.

### 3.8. Bread Volume and Height

The volume of the bread was determined by the rape seed displacement method according to the AACC 10–05 method [60]. The height of the loaves was measured using a digital caliper (Digi-MaxTM, SciencewareR, NJ, USA). The analyses were carried out in triplicate.

### 3.9. Crumb Porosity

Central bread slices from each loaf were visually compared with the eight Dallmann reference pictures [69] representing the cross section of breads with different crumb structures. The porosity of crumb was evaluated on the basis of the 8-degree Mohs scale as modified by Dallmann [70] where, for mold breads, 1 indicates non-uniform structure, large and irregular cells, and 8 indicates uniform compact structure, small and regular cells. The analyses were carried out in triplicate.

### 3.10. Determination of Bread Staling Rate

Bread was stored for 5 days at 25 °C, packed in cardboard. On the day of baking, 2 days and 4 days later, the loaf hardness and moisture were determined to evaluate the staling rate, according to the AACC method 10–10.03 [60]. Loaf hardness was assessed by using a texture analyzer (Z 0.5 Zwick Röell, Ulm, Germany) equipped with an 8 mm diameter cylindrical aluminum probe and configured with 50 g preload, at 100 mm/min preload speed, breaking point crust (force shut-down threshold) = 20%. The resulting peak force was measured in Newtons (N). Bread moisture was determined as reported in Section 3.4. The analyses were carried out in triplicate.

### 3.11. Statistical Analyses

Results are reported as mean ± standard deviation. The independent means were evaluated with the parametric ANOVA test and the Tukey test to determine the existence of significant differences (*p* < 0.001). The Statsoft Statistica 6 (Padova, Italy) software was used.

## 4. Conclusions

The reuse of artichoke processing waste has a recognized economic and environmental importance for the food industry, also considering its potential functional properties due to the richness of insoluble and soluble fibers. Our results show for the first time the feasibility of inserting artichoke by-products and side-streams into the value cycle of the durum wheat bread chain. This is a practical example of upcycling, and represents an important sustainability-oriented innovation practice contributing to waste reduction and efficient resource use.

Although the durum wheat breads with added artichoke waste flours had a lower porosity and volume than bread prepared from pure re-milled semolina, the same breads did not show relevant moisture losses during five days of storage. Some adjustments in the leavening phase and formulation, such as reducing leavening time to avoid dough collapsing, and increasing the hydration rate, would probably allow improvement in the volume and crumb porosity. It also has to be mentioned that the obtained results show that durum wheat bread, which is characterized by lower specific volume than soft wheat bread, is not the best candidate for being enriched with flours of artichoke waste, able to further reduce volume in a remarkable way due to their fiber content. Further investigations, however, are needed for achieving improved sensory features of the artichoke waste-enriched breads and for a full definition of their nutritional and functional features.

## Figures and Tables

**Figure 1 plants-11-03409-f001:**
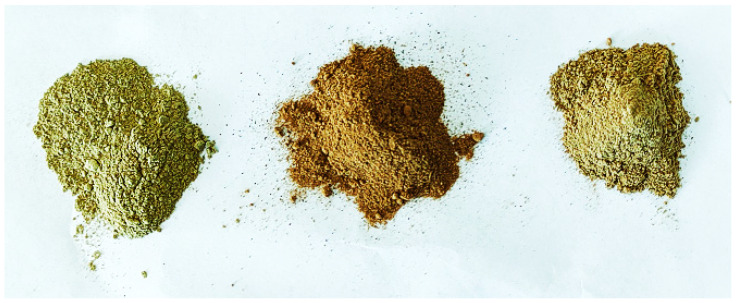
Flour prepared from artichoke waste. From left to right, flour of: artichoke bracts; artichoke stems; mix of artichoke bracts and stems 1:1.

**Figure 2 plants-11-03409-f002:**
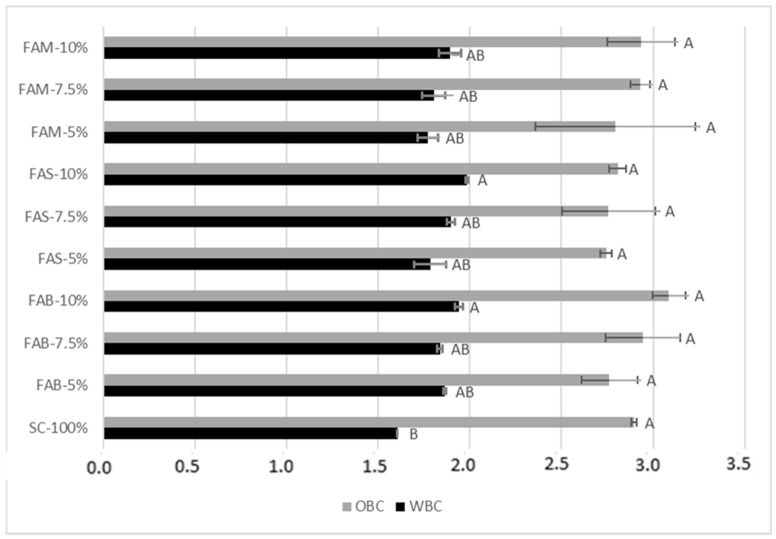
Water binding capacity (WBC; g water/g flour) and oil binding capacity (OBC; g oil/g flour) of re-milled semolina and of mixes prepared at increasing levels of replacement (5, 7.5, 10%) with flours from artichoke stems and bracts. SC-100% = re-milled semolina 100%, i.e., control; FAB = flour of artichoke bracts; FAS = flour of artichoke stems; FAM = flour of mixed artichoke bracts and stems. Different letters indicate a significant difference (*p* ≤ 0.001).

**Figure 3 plants-11-03409-f003:**
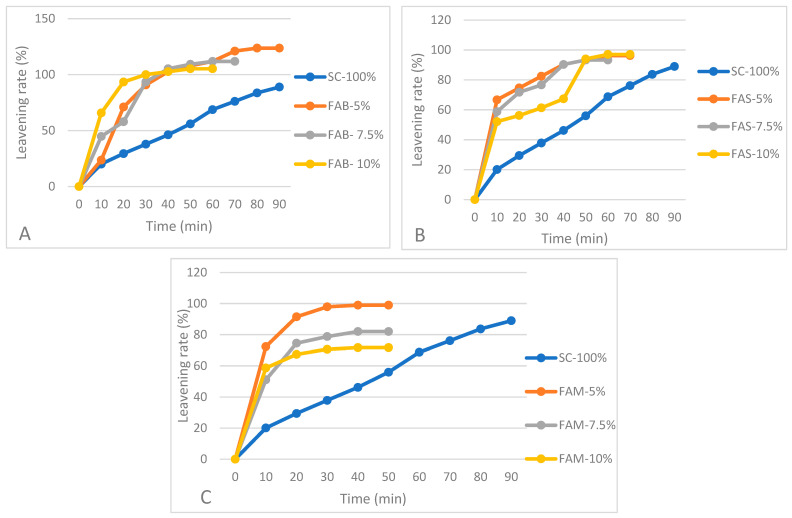
(**A**) Leavening rate (%) at increasing levels of replacement (5, 7.5, 10%) of re-milled semolina (SC-100%) with flour prepared from artichoke bracts (FAB). (**B**) Leavening rate (%) at increasing levels of replacement (5, 7.5, 10%) of re-milled semolina (SC-100%) with flour prepared from artichoke stems (FAS). (**C**) Leavening rate (%) at increasing levels of replacement (5, 7.5, 10%) of re-milled semolina (SC-100%) with flour of mixed artichoke bracts and stems (FAM).

**Figure 4 plants-11-03409-f004:**
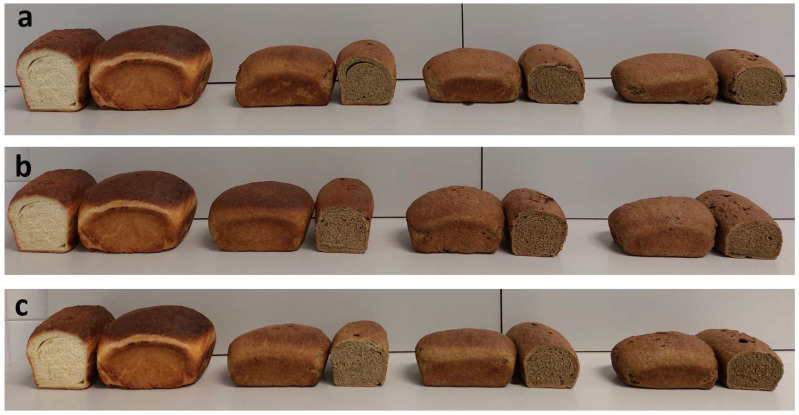
(**a**) Breads prepared, from left to right, with pure re-milled semolina and with flour mixes containing 5, 7.5 and 10% flours from artichoke bracts. (**b**) Breads prepared, from left to right, with pure re-milled semolina and with flour mixes containing 5, 7.5 and 10% flours from artichoke stems. (**c**) Breads prepared, from left to right, with pure re-milled semolina and with flour mixes containing 5, 7.5 and 10% flours from artichoke bracts and stems 1:1.

**Table 1 plants-11-03409-t001:** Moisture content of fresh artichoke stems and bracts and of the corresponding flour.

Sample	Moisture (g/100 g)
Fresh stems	67.0 ± 0.1
Fresh bracts	57.0 ± 0.1
FAS	6.0 ± 0.1
FAB	4.0 ± 0.1
FAM	5.0 ± 0.1

FAB = flour of artichoke bracts; FAS = flour of artichoke stems; FAM = flour of mixed artichoke bracts and stems.

**Table 2 plants-11-03409-t002:** Color parameters of re-milled semolina (control), of flours of artichoke stems and bracts, and of mixes prepared with re-milled semolina and artichoke flours at increasing levels of replacement (5, 7.5, 10%) of the former.

Type of Flour	Brown Index (100 − *L**)	*a**	*b**
*Pure flours*			
SC-100%	10.26 ± 0.01 i	−2.38 ± 0.00 k	17.17 ± 0.00 b
FAB-100%	34.56 ± 0.01 c	−1.21 ± 0.01 h	16.94 ± 0.01 c
FAS-100%	44.69 ± 0.01 a	2.68 ± 0.02 a	17.53 ± 0.01 a
FAM-100%	39.31 ± 0.04 b	0.71 ± 0.02 b	17.55 ± 0.01 a
*Mixes*			
FAB-5%	22.75 ± 0.01 g	−2.01 ± 0.01 j	15.27 ± 0.01 f
FAB-7.5%	22.72 ± 0.02 g	−1.93 ± 0.01 ij	15.55 ± 0.02 ce
FAB-10%	23.84 ± 0.01 fg	−1.87 ± 0.01 i	15.68 ± 0.01 d
FAS-5%	20.71 ± 0.01 h	−0.44 ± 0.03 e	15.09 ± 0.04 g
FAS-7.5%	24.15 ± 0.69 f	−0.11 ± 0.01 d	14.71 ± 0.01 i
FAS-10%	27.55 ± 0.02 d	0.34 ± 0.02 c	14.93 ± 0.04 h
FAM-5%	20.76 ± 0.01 h	−1.12 ± 0.01 h	15.09 ± 0.01 g
FAM-7.5%	23.59 ± 0.01 fg	−1.01 ± 0.01 g	15.35 ± 0.01 f
FAM-10%	26.20 ± 0.01 e	−0.78 ± 0.01 f	15.08 ± 0.01 g

SC-100% = re-milled semolina 100%, i.e., control; FAB = flour of artichoke bracts; FAS = flour of artichoke stems; FAM = flour of mixed artichoke bracts and stems. Different letters in a column indicate a significant difference (*p* ≤ 0.001).

**Table 3 plants-11-03409-t003:** Farinograph data of re-milled semolina (control) and of mixes prepared at increasing levels of replacement (5, 7.5, 10%) with flours from artichoke stems and bracts.

Type of Flour	Dough Development Time (min)	Stability (min)	Softening Degree (B.U.)	Water Absorption at 500 B.U. (g/100 g)
SC-100%	1.8 ± 0.1 h	3.2 ± 0.0 ef	55 ± 0.0 d	61.7 ± 0.1 e
FAB-5%	2.0 ± 0.1 gh	1.9 ± 0.1 f	34 ± 0.5 e	65.1 ± 0.1 d
FAB-7.5%	2.5 ± 0.1 fg	4.6 ± 0.1 e	67 ± 0.1 cd	65 ± 0.1 d
FAB-10%	2.8 ± 0.1 f	18.4 ± 0.0 a	141 ± 0.1 a	64.6 ± 0.1 d
FAS-5%	5.6 ± 0.1 e	7.2 ± 0.1 cd	76 ± 0.1 bc	66.1 ± 0.1 c
FAS-7.5%	6 ± 0.1 cd	5.9 ± 0.1 d	86 ± 0.1 b	68.8 ± 0.1 b
FAS-10%	7.3 ± 0.1 b	6.1 ± 0.1 d	83 ± 0.1 bc	70.2 ± 0.1 a
FAM-5%	5.9 ± 0.1 de	3.7 ± 0.1 ef	33 ± 0.2 e	66.4 ± 0.1 c
FAM-7.5%	6.5 ± 0.1 c	7.7 ± 0.1 c	66 ± 0.1 cd	68.6 ± 0.1 b
FAM-10%	3.9 ± 0.1 ed	15.1 ± 0.1 b	109 ± 0.1 ab	71.2 ± 0.1 a

B.U. = Brabender units. SC-100% = re-milled semolina 100%, i.e., control; FAB = flour of artichoke bracts; FAS = flour of artichoke stems; FAM = flour of mixed artichoke bracts and stems. Different letters in a column indicate a significant difference (*p* ≤ 0.001).

**Table 4 plants-11-03409-t004:** Mixograph and alveograph data of re-milled semolina (control) and of mixes prepared at increasing levels of replacement (5, 7.5, 10%) with flours from artichoke stems and bracts.

Type of Flour	Mixograph Data		Alveograph Data	
	Mixing Time (min)	Peak Height (M.U.)	W (10^−4^ × J)	P/L
SC-100%	2.65 ± 0.03 g	6.35 ± 0.03 d	225.5 ± 3.5 a	2.40 ± 0.03 e
FAB-5%	4.01 ± 0.01 cd	6.78 ± 0.01 c	170.0 ± 6.1 b	6.88 ± 0.43 cde
FAB-7.5%	3.53 ± 0.02 f	7.17 ± 0.02 b	101.0 ± 9.1 cde	10.87 ± 0.85 abc
FAB-10%	3.90 ± 0.02 de	7.35 ± 0.01 a	75.0 ± 6.4 de	15.20 ± 0.30 a
FAS-5%	4.48 ± 0.01 b	7.15 ± 0.02 b	129.0 ± 20.4 bc	7.00 ± 1.26 cde
FAS-7.5%	4.15 ± 0.02 c	6.79 ± 0.01 c	103.0 ± 18.9 cde	9.45 ± 2.38 bcd
FAS-10%	4.98 ± 0.01 a	6.71 ± 0.01 c	78.0 ± 7.7 de	14.37 ± 1.42 a
FAM-5%	2.49 ± 0.01 h	7.20 ± 0.01 b	115.0 ± 7.9 cd	4.76 ± 1.04 de
FAM-7.5%	3.54 ± 0.03 f	5.76 ± 0.03 f	105.0 ± 5.3 cde	5.83 ± 0.33 de
FAM-10%	3.78 ± 0.03 e	6.20 ± 0.03 e	61.0 ± 4.4 e	12.37 ± 0.57 ab

M.U. = mixograph units; W = dough strength; P/L = tenacity to extensibility ratio. SC-100% = re-milled semolina 100%, i.e., control; FAB = flour of artichoke bracts; FAS = flour of artichoke stems; FAM = flour of mixed artichoke bracts and stems. Different letters in a column indicate a significant difference (*p* ≤ 0.001).

**Table 5 plants-11-03409-t005:** Color of crust and crumb of durum wheat bread at increasing levels of replacement (5, 7.5, 10%) of re-milled semolina (control) with flours prepared from artichoke stems and bracts.

Type of Bread	Crust			Crumb		
	Brown Index (100 − *L**)	*a**	*b**	Brown Index (100 − *L**)	*a**	*b**
SC-100%	58.56 ± 3.03 abcd	16.38 ± 0.33 a	22.43 ± 0.81 bc	25.88 ± 0.01 l	−2.20 ± 0.03 h	24.11 ± 0.06 a
FAB-5%	48.09 ± 3.25 d	8.29 ± 0.10 bc	28.30 ± 1.14 a	42.95 ± 0.39 i	1.27 ± 0.28 e	20.41 ± 0.06 bcd
FAB-7.5%	48.37 ± 2.78 d	8.45 ± 0.65 bc	27.91 ± 0.47 ab	47.20 ± 0.35 fghi	2.02 ± 0.04 ef	19.46 ± 0.45 defgh
FAB-10%	56.93 ± 1.67 abcd	6.97 ± 0.74 c	21.93 ± 0.08 bc	55.67 ± 0.30 abc	2.46 ± 0.28 def	18.91 ± 0.16 gh
FAS-5%	51.89 ± 1.94 bcd	8.55 ± 5.10 bc	24.88 ± 2.28 abc	48.51 ± 0.54 efghi	2.47 ± 0.40 def	19.73 ± 0.20 cdefgh
FAS-7.5%	57.50 ± 0.21 abc	11.37 ± 1.39 abc	22.20 ± 2.97 bc	51.28 ± 0.18 cdefg	3.47 ± 0.39 abcd	20.62 ± 0.48 bc
FAS-10%	61.35 ± 1.64 a	10.14 ± 0.62 abc	19.83 ± 3.59 c	58.49 ± 0.06 a	3.95 ± 0.01 abc	18.83 ± 0.08 g
FAM-5%	54.21 ± 2.33 abcd	11.22 ± 0.11 bc	26.02 ± 0.23 ab	46.20 ± 0.07 hi	2.00 ± 0.02 ef	21.00 ± 0.03 b
FAM-7.5%	54.03 ± 0.98 abcd	10.60 ± 0.03 ab	25.60 ± 0.07 abc	52.13 ± 0.08 bcde	2.78 ± 0.14 cdef	20.17 ± 0.09 bcde
FAM-10%	57.85 ± 0.33 abc	10.07 ± 0.47 abc	22.21 ± 0.89 bc	56.63 ± 0.11 ab	3.14 ± 0.05 bcde	19.30 ± 0.03 efgh

SC-100% = re-milled semolina 100%, i.e., control; FAB = flour of artichoke bracts; FAS = flour of artichoke stems; FAM = flour of mixed artichoke bracts and stems. Different letters in a column indicate a significant difference (*p* ≤ 0.001).

**Table 6 plants-11-03409-t006:** Physical characteristics of durum wheat bread at increasing levels of replacement (5, 7.5, 10%) of re-milled semolina (control) with flours prepared from artichoke stems and bracts.

Type of Bread	Volume (cm^3^)	Weight (g)	Specific Volume (cm^3^/g)	Height (mm)	Porosity *
SC-100%	440.00 ± 3.54 a	144.50 ± 1.13 b	3.05 ± 0.05 a	81.9 ± 1.20 a	5.75 ± 0.35 b
FAB-5%	272.50 ± 3.54 e	153.03 ± 0.74 a	1.78 ± 0.03 d	61.2 ± 2.83 bcd	6.75 ± 0.35 ab
FAB-7.5%	232.50 ± 3.54 f	153.43 ± 0.18 a	1.52 ± 0.02 e	54.3 ± 1.20 cde	7.25 ± 0.35 ab
FAB-10%	210.00 ± 0.00 g	153.38 ± 0.95 a	1.37 ± 0.01 e	47.6 ± 2.26 e	7.75 ± 0.35 a
FAS-5%	366.25 ± 1.77 b	152.78 ± 1.10 a	2.40 ± 0.03 b	69.1 ± 1.13 b	6.00 ± 0.00 b
FAS-7.5%	316.25 ± 5.30 c	154.23 ± 0.32 a	2.05 ± 0.04 c	63.1 ± 1.34 bc	6.25 ± 0.35 ab
FAS-10%	292.50 ± 3.54 de	158.40 ± 0.07 a	1.85 ± 0.02 d	57.3 ± 0.35 cde	7.25 ± 0.35 ab
FAM-5%	302.50 ± 3.54 cd	153.98 ± 1.03 a	1.96 ± 0.01 cd	59.9 ± 0.00 bcd	6.25 ± 0.35 ab
FAM-7.5%	280.00 ± 0.00 e	156.80 ± 0.57 a	1.79 ± 0.01 d	54.3 ± 0.49 cde	7.00 ± 0.00 ab
FAM-10%	243.75 ± 1.77 f	157.63 ± 1.31 a	1.55 ± 0.02 e	52.0 ± 0.71 de	7.25 ± 0.35 ab

* Scale 1–8; 1 = non-uniform structure, large and irregular cells; 8 = uniform compact structure, small and regular cells. SC-100% = re-milled semolina 100%, i.e., control; FAB = flour of artichoke bracts; FAS = flour of artichoke stems; FAM = flour of mixed artichoke bracts and stems. Different letters in a column indicate a significant difference (*p* ≤ 0.001).

**Table 7 plants-11-03409-t007:** Variation of moisture content and hardness of durum wheat bread at increasing levels of replacement (5, 7.5, 10%) of re-milled semolina (control) with flours prepared from artichoke stems and bracts during 5 days of storage.

Days Elapsed from Baking	Type of Bread	Moisture (g/100 g)	Hardness (N)
**0**	SC-100%	31.2 ± 0.02 b(A)	12.1 ± 0.64 b(B)
	FAB-5%	32.8 ± 0.03 ab(A)	16.9 ± 0.94 ab(B)
	FAB-7.5%	34.4 ± 0.01 ab(A)	16.8 ± 0.54 ab(B)
	FAB-10%	33.9 ± 0.01 ab(A)	21.4 ± 0.43 a(B)
	FAS-5%	34.9 ± 0.00 ab(A)	12.2 ± 1.15 b(C)
	FAS-7.5%	35.0 ± 0.00 ab(A)	13.1 ± 1.86 ab(C)
	FAS-10%	36.7 ± 0.00 a(A)	11.9 ± 1.38 b(C)
	FAM-5%	35.9 ± 0.01 ab(A)	11.8 ± 2.18 b(C)
	FAM-7.5%	35.0 ± 0.01 ab(A)	11.4 ± 0.10 b(C)
	FAM-10%	37.1 ± 0.00 a(A)	13.3 ± 0.99 ab(C)
**2**	SC-100%	27.3 ± 0.01 b(AB)	33.7 ± 0.03 abc(A)
	FAB-5%	33.6 ± 0.01 a(A)	35.2 ± 0.80 ab(A)
	FAB-7.5%	33.3 ± 0.00 a(A)	37.1 ± 2.03 a(AB)
	FAB-10%	33.8 ± 0.01 a(A)	36.8 ± 0.13 a(AB)
	FAS-5%	32.8 ± 0.01 ab(A)	23.1 ± 0.41 d(B)
	FAS-7.5%	33.6 ± 0.00 a(A)	26.6 ± 1.85 bcd(B)
	FAS-10%	36.6 ± 0.01 a(A)	25.1 ± 0.36 cd(B)
	FAM-5%	34.9 ± 0.00 a(A)	24.6 ± 2.28 cd(B)
	FAM-7.5%	35.3 ± 0.01 a(A)	25.4 ± 1.80 cd(B)
	FAM-10%	35.0 ± 0.02 a(A)	32.93 ± 1.45 abc(B)
**4**	SC-100%	25.9 ± 0.00 b(B)	28.2 ± 0.24 e(A)
	FAB-5%	30.7 ± 0.03 ab(A)	44.3 ± 4.09 bcd(A)
	FAB-7.5%	31.3 ± 0.01 ab(A)	56.1 ± 0.45 ab(A)
	FAB-10%	30.6 ± 0.00 ab(A)	52.1 ± 1.91 abc(A)
	FAS-5%	25.2 ± 0.01 b(B)	40.3 ± 0.48 cd(A)
	FAS-7.5%	30.4 ± 0.01 ab(A)	36.9 ± 0.23 d(A)
	FAS-10%	34.9 ± 0.00 a(A)	44.9 ± 0.83 bcd(A)
	FAM-5%	30.1 ± 0.01 ab(B)	44.9 ± 0.97 bcd(A)
	FAM-7.5%	33.4 ± 0.00 a(A)	52.9 ± 1.14 abc(A)
	FAM-10%	31.0 ± 0.01 ab(B)	65.1 ± 3.58 a(A)

SC-100% = re-milled semolina 100%, i.e., control; FAB = flour of artichoke bracts; FAS = flour of artichoke stems; FAM = flour of mixed artichoke bracts and stems. Different lower case letters in a column indicate a significant difference (*p* ≤ 0.01) among different types of bread within the same day. Different capital letters in a column indicate a significant difference (*p* ≤ 0.01) for the same type of bread at different storage times.

**Table 8 plants-11-03409-t008:** Formulation of the experimental breads (g for 100 g of re-milled semolina).

Bread Type	Re-Milled Semolina	FAB	FAS	FAM	Yeast	NaCl	Ascorbic Acid	Sugar	Shortening	Water *
SC-100%	100	-	-	-	0.6	0.4	8 × 10^−4^	1.2	3.5	61.7
FAB-5%	95	5	-	-	0.6	0.4	8 × 10^−4^	1.2	3.5	65.1
FAB-7.5%	92.5	7.5	-	-	0.6	0.4	8 × 10^−4^	1.2	3.5	65.0
FAB-10%	90	10	-	-	0.6	0.4	8 × 10^−4^	1.2	3.5	64.6
FAS-5%	95	-	5	-	0.6	0.4	8 × 10^−4^	1.2	3.5	66.1
FAS-7.5%	92.5	-	7.5	-	0.6	0.4	8 × 10^−4^	1.2	3.5	68.8
FAS-10%	90	-	10	-	0.6	0.4	8 × 10^−4^	1.2	3.5	70.2
FAM-5%	95	-	-	5	0.6	0.4	8 × 10^−4^	1.2	3.5	66.4
FAM-7.5%	92.5	-	-	7.5	0.6	0.4	8 × 10^−4^	1.2	3.5	68.6
FAM-10%	90	-	-	10	0.6	0.4	8 × 10^−4^	1.2	3.5	71.2

SC-100% = re-milled semolina 100%, i.e., control; FAB = flour of artichoke bracts; FAS = flour of artichoke stems; FAM = flour of mixed artichoke bracts and stems. * Amount corresponding to the farinograph water absorption at 500 B.U.

## Data Availability

All available data are reported in the paper.

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
