# Peer review of "Waste from Artichoke Processing Industry: Reuse in Bread-Making and Evaluation of the Physico-Chemical Characteristics of the Final Product"

_plants, 2022, doi:10.3390/plants11243409_

Round 1
Reviewer 1 Report
The manuscript is fine as far as it goes—comprehensive and well referenced. However, I would have appreciated some sensory data and speculation about why the supplemented bread hardened faster than the control; the artichoke powder had high water-binding capacity, after all. Also—define “remilled” (semolina); and “Height” is spelled incorrectly in Table 6.
Author Response
The manuscript is fine as far as it goes—comprehensive and well referenced.
Answer: We thank the Reviewer for appreciating our work.
However, I would have appreciated some sensory data and speculation about why the supplemented bread
hardened faster than the control; the artichoke powder had high water-binding capacity, after all.
Answer: We agree with the Reviewer that a sensory evaluation would add value to the study, but at the moment we have not the possibility to perform such analysis. However, we are taking into account this suggestion and the sensory properties of the breads will be deepened in a future study. This study, indeed, has to be intended as a first work to establish the technological conditions for an optimal product development, in order to identify the artichoke flour giving the best results and the optimal percentage to be used. We will focus on the sensory and nutritional features in a subsequent study, in collaboration with
other teams.
A speculation about why the supplemented bread hardened faster than the control has been added (page 10, lines 279-283).
Also—define “remilled” (semolina); and “Height” is spelled incorrectly in Table 6.
Answer: Remilled has been defined (Page 2, lines 69-71) and height has been corrected in Table 6.

Reviewer 2 Report
I have reviewed the manuscript entitled " Waste from artichoke processing industry: Reuse in bread-making and evaluation of the physico-chemical characteristics of the final product. This work is well-presented and easy to read. But it lacks methodology from both the experimental design and the analytical point of view. The novelty and significance of this work should be made more explicit, especially in the abstract and conclusion sections. I recommended the publication of this work in Plants after major revisions.
Detailed remarks about the text are as follows:
Abstract: Abstract should be rewritten. Very general statements have been made. Important findings should be written in the study.
Page 1 line 23: It should be explained how the rates specified here are determined.
Sampling should be described. How did the authors detect the sampling date? How homogeneity and traceability were ensured and how many replicates were taken and analyzed.
The fact that the fiber ratio in the produced bread was not determined was seen as a deficiency.
I recommend performing sensory analysis with a trained sensory panel.
Conclusions should be more concise, with some considerations, for example, about the importance of the result, what they mean for the industry, which is the greatest novelties of the results, what future research in this field should be focused on, and similar.
Author Response
I have reviewed the manuscript entitled " Waste from artichoke processing industry: Reuse in breadmaking and evaluation of the physico-chemical characteristics of the final product. This work is well presented and easy to read. But it lacks methodology from both the experimental design and the analytical
point of view. The novelty and significance of this work should be made more explicit, especially in the abstract and conclusion sections. I recommended the publication of this work in Plants after major revisions.
Answer: The novelty and significance of the work have been made more explicit in the abstract.
Detailed remarks about the text are as follows:
Abstract: Abstract should be rewritten. Very general statements have been made. Important findings should be written in the study.
Answer: The abstract has been rewritten to make it more specific.
Page 1 line 23: It should be explained how the rates specified here are determined.
Answer: Thanks for suggesting. An explanation of how the rates were determined has been added (page 12, lines 312-316).
Sampling should be described. How did the authors detect the sampling date? How homogeneity and traceability were ensured and how many replicates were taken and analyzed.
Answer: Thanks for suggesting. An explanation of how the sampling date was determined has been added, as well as information on how homogeneity and traceability were ensured and how many replicates were taken and analyzed (page 11, lines 292-294; page 12, line 295-299,326).
The fact that the fiber ratio in the produced bread was not determined was seen as a deficiency.
Answer: We agree but this was out of the scope of the present article which was intended as a first work to establish the technological conditions for an optimal product development, in order to identify the artichoke flour giving the best results and the optimal percentage to be used. We will focus on the sensory
and nutritional features in a subsequent study, in collaboration with other teams.
I recommend performing sensory analysis with a trained sensory panel.
Answer: We agree with the Reviewer that a sensory evaluation would add value to the study, but at the moment we have not the possibility to perform such analysis. However, we are taking into account this suggestion and the sensory properties of the breads will be deepened in a future study. This study, indeed,
has to be intended as a first work to establish the technological conditions for an optimal product development, in order to identify the artichoke flour giving the best results and the optimal percentage to be used. We will focus on the sensory and nutritional features in a subsequent study, in collaboration with
other teams.
Conclusions should be more concise, with some considerations, for example, about the importance of the result, what they mean for the industry, which is the greatest novelties of the results, what future research in this field should be focused on, and similar.
Answer: The conclusions have been modified according to the Reviewer’s suggestions.

Round 2
Reviewer 2 Report
I checked again the revised paper entitled “Waste from artichoke processing industry: Reuse in bread-making and evaluation of the physico-chemical characteristics of the final product”. The general impression of the revised paper is that it has been improved by the authors.
The article can be accepted for publication in the Plants.
Best regards